# Post-COVID Symptoms in Occupational Cohorts: Effects on Health and Work Ability

**DOI:** 10.3390/ijerph20095638

**Published:** 2023-04-25

**Authors:** Nicola Magnavita, Gabriele Arnesano, Reparata Rosa Di Prinzio, Martina Gasbarri, Igor Meraglia, Marco Merella, Maria Eugenia Vacca

**Affiliations:** 1Post-Graduate School of Occupational Health, Università Cattolica del Sacro Cuore, 00168 Rome, Italy; nicolamagnavita@gmail.com (N.M.); gabrielearnesano93@gmail.com (G.A.);; 2Department of Woman, Child and Public Health, Fondazione Policlinico Universitario Agostino Gemelli IRCCS, 00168 Rome, Italy; 3Occupational Health Service, Local Health Unit Roma 4, 00053 Civitavecchia, Italy; 4Health Systems and Service Research, Post-Graduate School of Health Economics and Management, Università Cattolica del Sacro Cuore, 00168 Rome, Italy

**Keywords:** health promotion, fatigue, anosmia, stress, sleep, anxiety, depression, treatment, non-hospitalized, asymptomatic, long-COVID

## Abstract

Post-acute COVID-19 syndrome is frequently observed in workers and has a substantial impact on work ability. We conducted a health promotion program to identify cases of post-COVID syndrome, analyze the distribution of symptoms and their association with work ability. Of the 1422 workers who underwent routine medical examination in 2021, 1378 agreed to participate. Among the latter, 164 had contracted SARS-CoV-2 and 115 (70% of those who were infected) had persistent symptoms. A cluster analysis showed that most of the post-COVID syndrome cases were characterized by sensory disturbances (anosmia and dysgeusia) and fatigue (weakness, fatigability, tiredness). In one-fifth of these cases, additional symptoms included dyspnea, tachycardia, headache, sleep disturbances, anxiety, and muscle aches. Workers with post-COVID were found to have poorer quality sleep, increased fatigue, anxiety, depression, and decreased work ability compared with workers whose symptoms had rapidly disappeared. It is important for the occupational physician to diagnose post-COVID syndrome in the workplace since this condition may require a temporary reduction in work tasks and supportive treatment.

## 1. Introduction

The SARS-CoV-2 virus can affect numerous organs and systems, inducing the multi-organ syndrome known as COVID-19 [1]. As occurs in a number of viral infections, for example EBV, HSV and HTLV [2], after the acute phase, symptoms may persist for many weeks, or previous chronic morbid conditions may be exacerbated. This clinical picture is called post-acute COVID-19 syndrome [3], or long-COVID. To date, there are no fixed criteria for the diagnosis of ‘long-COVID’ [4], although this illness is known to include a wide spectrum of clinical manifestations that may imply underlying multi-organ disorders. A provisional definition could be symptoms and potential sequelae that continue to persist at four weeks from onset [5]. Since these ongoing symptoms may combine, some patterns have emerged: Yong and Liu observed six different clinical pictures: non-severe COVID-19 multi-organ sequelae, pulmonary fibrosis sequelae, myalgic encephalomyelitis or chronic fatigue syndrome, postural orthostatic tachycardia syndrome, post-intensive care syndrome and medical or clinical sequelae [6].

The experience of returning to work after SARS-CoV-2 infection is a highly relevant topic for occupational health. Studies performed on workers during the first phases of the pandemic suggest that long recovery times may be related to high severity [7], but they may also be a consequence of post-COVID [8] since long-lasting syndromes have also been reported independently of acute phase severity, hospitalization, and treatment [3,9]. The prevalence of post-COVID cases is higher in hospitalized patients than in cases not requiring hospitalization [10]. Studies on serious patients discharged from hospital show that between 33% and 87% had post-COVID syndrome [1,11,12,13,14,15,16]. In non-hospitalized cohorts, up to 67% reported long-COVID-19 [17,18].

A systematic literature search involving more than 735,000 cases up to January 2022, revealed that 45% of COVID-19 survivors, regardless of hospitalization status, were experiencing a range of unresolved symptoms at ∼4 months [19]. Prevalence was, however, higher in hospitalized than in non-hospitalized patients (52.6% vs. 34.5%) [19]. According to a meta-analysis conducted by the Global Burden of Disease Long COVID Collaborators on 1.2 million individuals who had symptomatic SARS-CoV-2 infection in 2020 and 2021, an estimated 6.2% experienced long-COVID symptom clusters [20]. The estimated mean long-COVID symptom cluster duration was 9.0 months (95% UI: 7.0–12.0 months) among hospitalized individuals, and 4.0 months (95% UI: 3.6–4.6 months) among non-hospitalized individuals. Among individuals with long-COVID symptoms 3 months after symptomatic SARS-CoV-2 infection, an estimated 15.1% (95% UI: 10.3–21.1%) continued to experience symptoms at 12 months [20].

As of 13 March 2023, COVID-19 had infected 681 million individuals worldwide (https://www.worldometers.info/coronavirus/ accessed on 13 March 2023). Based on a cautious estimate of 6.2% survivors who experience persistent symptoms, this means over 42,200,000 individuals might be affected by the long-term consequences of COVID-19. Should this virus continue to circulate among us for years to come, the long-term effects could increase exponentially [21].

While the long-term implications of the multi-organ manifestations of COVID-19 are now well documented, the potential occupational impact of these manifestations still requires investigation.

The purpose of this study, conducted by the Catholic University of the Sacred Heart in 2020 and 2021 during the first four waves of COVID-19 in Italy [22], was to evaluate the incidence of SARS-CoV-2 infection in workers and ascertain which association of symptoms was more frequent. Moreover, the study aimed to determine whether protracted symptoms were associated with occupational stress, sleep disturbances or mental health.

## 2. Materials and Methods

### 2.1. Population and Type of the Study

In Italy, employees who are exposed to occupational hazards have an annual medical check-up in the workplace to assess their occupational fitness. Workers were asked to complete a questionnaire as they awaited their medical examination. Participation in the survey was optional. The occupational physician reviewed the questionnaire during the subsequent physical examination and was able to delve deeper into anamnestic information and, if necessary, send the employee to National Health Service (NHS) institutions for diagnostic procedures or therapy.

This study was cross-sectional and retrospective. Based on the data collected, the population was divided into three subgroups: (i) workers who had contracted the SARS-CoV-2 infection between the start of the pandemic (in Italy, February 2020) and the date of the medical visit (during 2021), and had passed the acute phase without continuing symptoms; (ii) workers who had symptoms 4 weeks after the acute phase (post-COVID syndrome cases); and (iii) workers who had not contracted the infection.

The study received the approval of the Lazio 1 Ethics Committee (project number 1343, 20 October 2021) and the approval of the Ethics Committee of the Catholic University of the Sacred Heart (project number 4079, approved 4 November 2021).

The data were deposited in a publicly available database.

### 2.2. Questionnaire

The workers were administered a questionnaire designed to investigate the occurrence of SARS-CoV-2 infection, the duration and severity of symptoms and the possible presence of post-COVID syndrome. The persistent symptoms investigated were: headache, insomnia, tiredness, muscle aches, weakness, anosmia, breathlessness, dysgeusia, fatigability, malaise, tachycardia, dyspnea, anxiety, depression, brain fog, and memory loss. Workers were also given additional questionnaires to evaluate factors potentially associated with the syndrome.

A short Italian version [23,24] of the Effort/Reward Imbalance Questionnaire [25] was used to measure occupational stress. According to this model, being exposed to a frequent lack of reciprocity at work can increase the possibility of developing incident stress-related disorders. All items have graded responses on a 4-point Likert scale, so the resulting subscales range from 3 to 12 (effort) and from 7 to 28 (reward), respectively. The shortened version of the questionnaire contains three questions about the effort variable and seven about the reward variable. The weighted ratio between the two variables, effort/reward imbalance (ERI), indicates a state of distress if values are greater than one. In this study, the test score reliability coefficient (Cronbach’s alpha) for the ERI effort subscale was 0.777, while Cronbach’s alpha for the reward subscale was 0.725.

The Italian version [26] of the Pittsburgh Sleep Quality Index (PSQI) [27], which consists of 18 items, was used to evaluate the quality of sleep. Poor sleep quality is indicated by a score of 5 or above (bad sleeper). In this research, Cronbach’s alpha was 0.833.

Sleepiness was evaluated using the Epworth Sleepiness Scale ESS [28] consisting of 8 questions that rated, on a 4-point scale (0–3), the chances of dozing off or falling asleep while engaged in eight different activities. An ESS score > 10 indicated excessive daytime sleepiness (EDS). In our sample, the internal consistency measured by Cronbach’s alpha value was 0.666.

Fatigue was measured with the Fatigue Assessment Scale [29] composed of 10 items rated on a 5-point scale; the final score ranges from 10 to 50, with higher scores indicating greater fatigue. A score equal to or higher than 24 has been proposed as a cut-off for classifying fatigue on the FAS [30]. In this study, Cronbach’s alpha was 0.822.

The Italian version [31] of the Goldberg Anxiety and Depression Scale (GADS), consisting of an 18-item self-report list of symptoms established specifically to assess the likelihood of the onset of anxiety or depression, was used to measure both anxiety and depression. Nine binary questions make up each subscale, and a point is given for each affirmative response. An anxiety subscale score of 5 or more and a depression subscale score of 2 or more both suggest possible clinically recognizable anxiety or depression, respectively [32]. In this study, the GADS anxiety subscale had a Cronbach’s alpha of 0.858 and the value for the GADS depression subscale was 0.785.

Happiness was measured using the Abdel-Khalek single item (“Do you feel happy in general?”) rated on an 11-point scale (0–10) [33].

Current work ability compared with lifetime best was measured using the first item of the Work Ability Index (WAI) [34], rated on a 11-point scale (0–10).

### 2.3. Statistics

The variables were described in terms of mean and standard deviation and checked for normality, using the Kolmogorov–Smirnov and Shapiro–Wilk tests. The issue of normality was not critical in this study, because, as reported by Lumley et al. [35], the assumption of normality is only required for small samples, due to the central limit theorem. With sample sizes exceeding 30, as it is the case here, violations of the normality assumptions are not problematic, and they become less and less problematic, even when extreme, as the sample size increases. Comparisons between the means of the variables were made using Student’s t-test or Mann–Whitney’s U-test, the latter for comparing ordinal variables. Comparisons between proportions were made using Pearson’s chi-square test. We used k-means cluster analysis to identify groups of patients who had had similar post-COVID symptoms so that each observation belonged to the cluster with the closest mean (cluster centers or cluster centroid). Convergence was defined as no or small change in cluster centers after an iterative calculation.

A linear regression model was used to evaluate the effect of individual and occupational variables on the number of post-COVID symptoms.

The IBM Corp. Released 2019. IBM SPSS Statistics for Windows, Version 26.0. Armonk, NY, USA: IBM Corp was used for the analyses.

## 3. Results

We examined 1378 individuals (males 462, 33.5%; mean age 48.71 ± 11.19). Most were employed in hospitals or healthcare establishments (814, 59.1%), while the remainder came from trade, industry, and social services (564, 40.9%). A total of 1422 individuals were examined in 2021; the participation rate was, therefore, 96.9%.

One hundred and sixty-four of these individuals (11.9%) contracted COVID-19 in 2020–2021. The prevalence of infection was significantly higher in females (123 cases, 75%) than in males (41, 25%) (Pearson chi square = 6.07, *p* = 0.014). No significant difference in age was found between those who contracted the disease (49.74 ± 10.68 years) and those who failed to contract it (48.57 ± 11.25) (Student’s t = 0.90, *p* = 0.36). 74.4% of all registered COVID-19 cases occurred among hospital workers. This incidence of COVID-19 was significantly higher than in other workers: 122 cases occurred among hospital workers (15%), while only 42 cases (7.2%) were reported among the remaining share of the sample (Pearson chi square = 18.07, *p* < 0.001). The difference was most noticeable during the first wave of infection when 90% of cases that occurred before June 2020 (69 out of 77) affected hospital workers.

Most infections occurred before June 2020 (first pandemic wave, 77 cases, 47.2%). Only a few workers reported being infected with SAS-CoV-2 in the summer between June and September 2020 (7 cases 4.3%), while there were 79 cases of infection (48.5%) during subsequent pandemic waves after October 2020. The course of the disease was asymptomatic in 27 workers (16.6%), mild and without the need for treatment in 65 cases (39.9%), and of moderate severity in 63 individuals (38.7%) who were treated at home. Only eight workers (4.9%) had to resort to hospital treatment. In more than half of the total number of cases, the duration of the infection was very short (less than a week (31, 19.0%)) or short (between 8 and 15 days (56, 34.4%)). In 45 cases, the acute phase lasted between 16 and 21 days (27.6%), and in 31 cases over three weeks (19.0%).

Workers who had been affected by COVID-19 reported significantly higher fatigue and a higher risk of depression compared with controls. In the other parameters, no significant difference was found between infected and non-infected workers (Table 1).

One hundred and fifteen individuals (70% of those who became infected, 8.3% of the sample) reported one or more symptoms persisting after the acute phase. The most frequent symptoms were tiredness, anosmia, dysgeusia, fatigability, muscle aches, and headache (Table 2). Post-COVID symptoms were significantly more frequent in females (75.6%) than in males (53.7%) (Pearson’s chi square *p* = 0.008). We observed no difference in the age of those who had persistent symptoms (49.98 ± 9.93) or in those who recovered without persistent symptoms (49.16 ± 12.34) (Student’s t, *p* = 0.655). The prevalence of post-COVID complaints increased in relation to the severity of the acute form: it was present in 37% of asymptomatic, 72% of mild cases, 81% of cases with home treatment, and 88% of hospitalized workers (chi square *p* < 0.0001). The duration of the acute phase also had a relationship with the occurrence of post-COVID symptoms. They were present in 55 % of infections lasting less than one week, 64 percent of acute forms lasting up to two weeks, 76 percent of cases lasting up to three weeks, and 90 percent of longer forms (chi square *p* < 0.05). The number of symptoms manifested by each post-COVID worker was significantly correlated with the reduction in work ability measured using the Work Ability Index (Spearman’s correlation coefficient = −0.418 *p* < 0.001).

Moreover, 18 individuals (1.3% of the population, i.e., 11% of those who contracted the infection) reported worsening of the morbid conditions from which they had previously suffered. Since the health surveillance medical examinations were not motivated by the onset of the disease or by the need to check the state of health before returning to work (a task entrusted to the NHS in Italy), they were performed at variable intervals following the acute phase. During medical examination in the workplace, 26 workers (2.6% of the total, 16% of the COVID cases) reported currently suffering from symptoms linked to the previous infection.

To verify the hypothesis that the higher levels of fatigue and depression perceived by workers who had had COVID-19 were due to the post-COVID syndrome, we repeated the comparisons excluding those who had reported protracted symptoms. No demonstrable difference was observed between the survivors with post-COVID syndrome and the other workers (Table 1, last column).

Cluster analysis enabled us to observe three different post-COVID symptom patterns. In the iterative calculation, convergence due to no or small changes in cluster centers was achieved after seven iterations. The most prevalent picture (47 cases) was characterized by anosmia and dysgeusia. A numerically similar group of workers (44 cases) reported fatigability, tiredness, and weakness. A smaller number of workers (24 cases—about one-fifth of the total) reported a heterogeneous combination of symptoms, including headache, insomnia, muscle aches, dyspnea, tachycardia, malaise and anxiety, in addition to anosmia, weakness, tiredness, and fatigability (Table 3).

Workers who had symptoms more than 4 weeks after infection (post-COVID) were found to have poorer quality sleep, increased fatigue, anxiety, depression, and decreased work ability compared with workers whose symptoms had rapidly disappeared (Table 4).

In a multiple linear regression model, the number of persistent symptoms in workers was significantly associated with anxiety and occupational stress (Table 5). Cases of distress among workers with post-COVID syndrome had the same prevalence as in the other members of the group (31.1% vs. 30.1%). Thirty-five (30.4%) workers with post-COVID were at risk of being affected by clinical anxiety, while prevalence in the rest of the sample was 18.3% (*p* < 0.002). Among the workers with post-COVID syndrome, there were 54 cases of depression (47%) compared with an average prevalence of 31.5% (*p* < 0.001). Sleep problems in post-COVID were more prevalent than in the rest of the group (42.5% in post-COVID, 33.8% in others), but the difference was not significant.

## 4. Discussion

This study, which, to the best of our knowledge, is the only one performed in the workplace with interviews and medical examinations on all active workers, made it possible to evaluate the prevalence of post-COVID syndrome and its impact on workers’ health and work ability.

In this heterogeneous case study conducted by the Rome Catholic University of the Sacred Heart on workers exposed to occupational risks in different companies, almost 12% contracted the SARS-CoV-2 infection during the first pandemic waves between 2020 and 2021. The incidence of infection in healthcare workers was much higher than in other occupational sectors, especially at the beginning of the epidemic. The high prevalence of infection in healthcare personnel is described in the literature. Studies conducted in 2020 enabled researchers to estimate an 11% (CI: 7–15) pooled prevalence of SARS-CoV-2 infection in healthcare workers [36]. In 2022, our own longitudinal study of anesthetists in one of two hub-COVID-19 hospitals in central Italy, continuously and exclusively exposed to high risk of infection, found a 20% incidence after the first four pandemic phases; three-quarters of these infections had occurred during the first wave [37]. Another early meta-analysis estimated an overall incidence of 10.1% (95% CI: 5.3–14.9) of COVID-19 among HCW [38]. The incidence obviously increased as the pandemic continued. A later meta-analysis estimated a positivity rate equal to 51.7% (95% CI: 34.7–68.2), with a prevalence of hospitalization of 15.1% and mortality of 1.5% [39].

A general unpreparedness, the lack of personal protective equipment and safety procedures, difficulty in promptly ascertaining the existence of an infection before the development and dissemination of rapid tests, and the guidelines that compelled workers to remain on duty if their temperature did not exceed 37.5 °C may have been among the causal factors. Recent studies have shown that in Italy, one in five people who tested positive for SARS-CoV-2 during the first epidemic wave of COVID-19 (from 29 February 2020 to 7 July 2020) were asymptomatic [40]. In our study of healthcare workers during the first pandemic wave in spring 2020, one in three of the HCWs who tested positive never manifested symptoms [41]. In the aforementioned meta-analysis, 40% (95% CI: 17–65) of the HCWs who tested positive for COVID-19 by reverse transcription-polymerase chain reaction were asymptomatic at the time of diagnosis [36]. While public health experts are debating the question of whether asymptomatic and late spreaders can continue virus transmission in the community, there is little doubt that keeping untested workers on duty after unprotected exposure was a flaw in the healthcare system.

Although cases were mostly mild in this occupational cohort, 70% of workers who contracted SARS-CoV-2 in the early stages of the pandemic experienced symptoms that persisted for more than four weeks after the acute phase. In four-fifths of the cases the ongoing symptoms consisted of anosmia/dysgeusia complex or fatigue, tiredness, and weakness. One-fifth of cases reported a heterogeneous combination of symptoms, with a frequent reactivation of pre-existing conditions. In our observations, as in the literature [42,43], the frequency of post-COVID symptoms was correlated with the duration and severity of the acute phase.

The distribution of symptoms in our cohort was consistent with what has been reported in the literature. In our sample we observed a high frequency of anosmia and dysgeusia—two symptoms that often appear together to form a recurrent post-COVID picture. Anosmia, i.e., the loss of or change in sense of smell, is one of the most common symptoms in the acute phase of COVID-19 and the post-COVID syndrome [44]. Chemosensory deficits are often the earliest, and sometimes the only signs of the SARS-CoV-2 virus in otherwise asymptomatic carriers [45]. SARS-CoV-2 is a neurotropic virus that can pass from the olfactory epithelium in the nose to infect the brain [46]. Anosmia and dysgeusia are among the most frequently reported post-COVID symptoms in children, adolescents [47] and hospitalized adults [16]. In severe cases, post-COVID-19 neurological sequelae include persistent symptoms of anosmia, dysgeusia, headache, fatigue, myalgia, and sleep disturbance [48].

In our cohort, fatigue, tiredness, and weakness formed a frequent triad of post-COVID symptoms. This finding confirmed previous observations. A meta-analysis of studies on hospitalized and non-hospitalized patients indicated that fatigue, dyspnea, and sleep disturbance are the most common post-COVID symptoms and may be experienced up to 12 months after the acute phase [49]. According to a systematic literature review, fatigue affects more than 50% of patients [50]. Of the numerous post-COVID symptoms, fatigue and dyspnea are the only ones significantly associated with acute illness severity [51]. The neuropsychological assessment of post-COVID patients showed that fatigue and cognitive dysfunction were the most frequent symptoms associated with loss of work capacity [52].

In a few cases, in addition to fatigue and sensory disturbances, workers experienced dyspnea, tachycardia, muscle aches, headache, insomnia, malaise, and anxiety. These workers often reported the reactivation of previous pathologies and had the most evident reduction in work ability. The pattern of symptoms observed in these workers was similar to that observed in patients hospitalized with severe forms of the virus. When compared with patients discharged from hospital, only dyspnea frequency was lower in these subjects affected by a mild form of the virus. Studies have shown that post-COVID can affect both individuals who have the severest forms of the disease and those who have minor manifestations of the acute illness, although the latter have a lower frequency of this syndrome, as also shown in our study. Post-COVID can affect numerous organs and systems in a similar way to COVID-19 [53]. However, a significant deterioration in the quality of life was observed only in patients who had been admitted to an ICU [54].

In our study, post-COVID workers showed a greater risk of insomnia, anxiety, and depression than their colleagues. Interestingly, this observation enabled us to recognize the etiopathogenetic role of SARS-CoV-2 on mental health, thereby distinguishing it from the action of many other occupational stressors which are common during epidemics. Numerous studies have reported a high prevalence of anxiety and depression in health-care workers during the pandemic; a meta-analysis estimated a pooled prevalence of 30.0% for anxiety (95% CI, 24.2–37.05); 31.1% for depression (95% CI, 25.7–36.8), and 56.5% for acute stress (95% CI—30.6–80.5) [55]. It is not easy to interpret these data since they are taken from cross-sectional studies, mostly without a control group, and do not contain information on the prior condition of the workers. Moreover, these studies do not enable us to understand how many of the problems observed can be attributed to the disease and how many to the type of work. Through longitudinal studies we were able to observe how the mental health status of frontline workers varied in relation to occupational stressors associated with the different phases of the pandemic [37], while a study conducted during the first phase of the pandemic showed that the rate of anxiety, depression and sleep disturbances is three to four times higher in individuals who contracted the disease than in subjects not infected [41]. The same study showed that workers who had had COVID-19 have worse sleep quality and a higher risk of anxiety and depression than others. This difference is entirely due to the fact that many of the sufferers developed post-COVID syndrome, since no difference in mental health levels was detected between workers who survived the disease without sequelae and those who were not affected by it. Anxiety, depression, poor quality sleep and fatigue were significantly greater in post-COVID workers than in others.

Our study revealed an association between occupational stress and anxiety and the number of symptoms in post-COVID syndrome. Since it was a cross-sectional study, we cannot attribute a definite interpretation to this association. We can assume that stress and anxiety produce a greater number of disturbances or push workers to report a greater number of symptoms. However, the opposite hypothesis, i.e., that workers with more articulated and complex symptoms become anxious and stressed, cannot be ruled out. Longitudinal studies based on further medical examinations in the workplace will indicate which hypothesis is correct.

The workers who were the subject of this study were mainly employed in health services. Healthcare workers were the most exposed to the first wave of the pandemic and were consequently the most studied, also with reference to post-COVID symptoms.

A telephone survey conducted three months after infection among 427 health care workers from a hospital in central Italy showed that 33.8% had persistent symptoms, consisting mainly of widespread pain, changes in taste and smell, and asthenia. In addition, 37.2% reported insomnia, daytime sleepiness, or both [56]. In this case study, the authors observed that the women had a greater tendency to report post-COVID symptoms, as in our experience. They also reported an association with age, which could be due to the fact that the telephone interviews investigated pain, a symptom not pathognomonic of post-COVID syndrome and known to be associated with age.

A survey monkey report of a sample of British healthcare workers seropositive for SARS-CoV-2 anti–spike IgG during the first pandemic wave, to which slightly more than one-tenth of those affected responded, indicated that fatigue, shortness of breath and sleep disturbances were the most common symptoms of long-COVID [57]. A study of healthcare workers at a Stockholm hospital who were seropositive for SARS-CoV-2 anti-spike IgG conducted via a smartphone app observed that they reported more frequent anosmia, fatigue, ageusia, and dyspnea than workers with negative serology [58]. A Swiss longitudinal study with weekly electronic questionnaires involving 3334 workers from 23 hospitals, 17% of whom had a positive nasopharyngeal swab test, observed an increase in long-COVID symptoms only in cases with a positive swab, but not in workers with positive serology. In this study, persistent symptoms were more frequent in young people, and in women. The most common symptoms of long-COVID were exhaustion/burnout, weakness/tiredness, and impaired taste/olfaction. The marked reduction in the number of participants over the course of the study limited the reliability of psychometric test results; however, the authors observed that physical activity at baseline was negatively associated with neurocognitive impairment and fatigue scores [59]. Taken together, these studies observed that a proportion of workers who contracted the SARS-CoV-2 infection had long-lasting symptoms, such as fatigue, taste and smell disorders and sleep disturbances. Women were more sensitive, while age did not show a clear association with post-COVID symptoms. Our study confirmed these observations and furthermore evaluated the impact of the post-COVID syndrome on anxiety, depression and quality of sleep, fatigue, sleepiness and workability. These findings should motivate targeted prevention interventions in the workplace.

### 4.1. Future Perspectives

Doctors in charge of workers’ health surveillance have been reporting very severe cases of post-COVID syndrome to management to obtain a temporary reduction in workload or night work which may interfere with symptoms. Moreover, occupational doctors have also referred workers experiencing post-COVID syndrome to their general practitioner (GP) or NHS facilities for treatment. Recent studies have shown that daily supplements based on amino acids, minerals, vitamins, and plant extracts may improve fatigue and muscle function [60]. Other treatment options include molecular hydrogen inhalation, hyperbaric oxygen therapy, aerobic training, strengthening exercises, diaphragmatic breathing techniques, and mindfulness training [61]. Several treatments have been proposed for the olfactory consequences of the infection; to date, however, there is very limited evidence available concerning the efficacy and potentially negative effects of treatment for persistent olfactory dysfunction resulting from COVID-19 infection [62]. Other therapeutic approaches may be useful in chronic inflammation, metabolic alterations, endothelial dysfunction, and gut dysbiosis that concur in the pathogenesis of post-COVID syndrome [63]. Rehabilitation could improve dyspnea, anxiety, and the quality of life [64].

### 4.2. Limitations

This study has some limitations. The first results from the use of a convenience sample consisting of workers subjected to surveillance in 2021. Since this sample included only some professional categories, it cannot be extrapolated to encompass all workers. However, the high level of participation of workers makes this study a census. The fixed schedule of medical examinations during which promotional intervention was proposed prevented us from examining workers at the same interval after infection. However, this procedure is envisaged in occupational medicine to eliminate the costs of health promotion and has proved effective even in the absence of funding. Although the cross-sectional nature of the study precludes recognition of causality, the continuation of health surveillance in the workplace will make it possible to detect the evolution of symptoms and the relationship between the syndrome and psychosocial variables.

## 5. Conclusions

This study revealed that about three-quarters of workers who were infected with SARS-CoV-2 experienced protracted neurosensory symptoms (anosmia and dysgeusia, fatigability, weakness, and tiredness) sometimes associated with dyspnea, tachycardia, headache, sleep disturbances, anxiety, and depression. Post-COVID syndrome is associated with reduced work ability. A temporary reduction in workload and the elimination of night shifts have been widely recommended for facilitating convalescence.

## Figures and Tables

**Table 1 ijerph-20-05638-t001:** Comparison between workers affected by-COVID-19 (164 cases) and controls.

Variable	COVID-19 Cases Mean ± s.d.	Controls Mean ± s.d.	Mann–Whitney U *p*	Mann–Whitney U ^1^ *p*
Stress	0.87 ± 0.37	0.85 ± 0.44	0.155	0.259
Anxiety	2.58 ± 2.74	2.19 ± 2.62	0.069	0.099
Depression	1.78 ± 2.06	1.40 ± 1.99	0.005	0.330
Low sleep quality	5.50 ± 3.87	4.80 ± 3.31	0.056	0.396
Daytime sleepiness	5.14 ± 3.85	4.78 ± 3.24	0.527	0.833
Fatigue	19.10 ± 5.79	18.16 ± 5.66	0.022	0.343
Happiness	7.44 ± 1.83	7.39 ± 3.37	0.133	0.365
Work Ability	8.78 ± 1.87	8.75 ± 1.90	0.975	0.611

Note: s.d.: standard deviation, ^1^ = Excluding post-COVID cases.

**Table 2 ijerph-20-05638-t002:** Frequency of post-COVID symptoms.

Symptom ^1^	Post-COVID Cases	Prevalence ^2^
Tiredness	77	47.0
Anosmia	63	38.4
Weakness	59	36.0
Dysgeusia	53	31.7
Fatigability	48	29.3
Muscle aches	47	28.7
Headache	45	27.4
Malaise	37	22.6
Insomnia	36	22.0
Tachycardia	33	20.1
Dyspnea	31	18.9
Anxiety	28	17.1
Breathlessness	16	9.8
Depression	15	9.1
Brain fog	12	7.3
Memory loss	11	6.7

^1^ Symptoms reported beyond four weeks from onset; ^2^ referred to all infected workers.

**Table 3 ijerph-20-05638-t003:** Clustering of post-COVID symptoms.

Symptom ^1^	Cluster 1 (47 Cases)	Cluster 2 (44 Cases)	Cluster 3 (24 Cases)
Anosmia	1		1
Dysgeusia	1		
Weakness		1	1
Tiredness		1	1
Fatigability		1	1
Muscle aches			1
Headache			1
Insomnia			1
Dyspnea			1
Tachycardia			1
Malaise			1
Anxiety			1
Breathlessness			
Depression			
Brain fog			
Memory loss			

^1^ Symptoms persisting after four weeks from onset.

**Table 4 ijerph-20-05638-t004:** Comparison between workers with post-COVID syndrome and workers not reporting persistent symptoms.

Variable	Post-COVID Cases (*n* = 115) Mean ± s.d.	Healed Workers (*n* = 49) Mean ± s.d.	Mann–Whitney U *p*
Stress	0.86 ± 0.37	0.89 ± 0.37	0.745
Anxiety	3.00 ± 2.83	1.59 ± 2.26	0.001
Depression	2.07 ± 2.13	1.10 ± 1.74	0.002
Low sleep quality	5.93 ± 4.00	4.48 ± 3.38	0.027
Daytime sleepiness	5.26 ± 4.09	4.88 ± 3.29	0.825
Fatigue	19.95 ± 6.09	17.13 ± 4.50	0.007
Happiness	7.47 ± 1.79	7.37 ± 1.95	0.963
Work ability	8.56 ± 2.01	9.31 ± 1.36	0.012

Note: s.d.: standard deviation.

**Table 5 ijerph-20-05638-t005:** Linear regression model assessing the effect of individual and occupational variables on the number of post-COVID symptoms.

	Standardized Coefficient Beta	t	*p*
Age	0.024	0.320	0.749
Gender	0.107	1.391	0.167
ERI	−0.211	−2.565	0.011
Anxiety	0.337	2.828	0.005
Depression	0.111	0.973	0.332
Sleep quality	0.072	0.641	0.523

## Data Availability

Data are freely available in Zenodo.

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
