# Peer review of "Post-COVID Symptoms in Occupational Cohorts: Effects on Health and Work Ability"

_ijerph, 2023, doi:10.3390/ijerph20095638_

Round 1

Reviewer 1 Report

ijerph-2346702: "Post-Covid symptoms in occupational cohorts: effects on health and work ability" about the COVID-19 pandemic has had a significant impact on the health and workability of many occupational cohorts. As the pandemic goes on, people are becoming more worried that people who have been infected with the virus could have long-term health effects, such as post-COVID symptoms. Post-COVID symptoms can include fatigue, shortness of breath, chest pain, and other respiratory symptoms. These symptoms can have a significant impact on an individual’s ability to perform their job duties, as well as their overall health.

It’s very good research work, but there is still a weak manuscript; please improve it and follow my suggestions for clear readers.

An abstract is a short summary of a manuscript that gives an overview of the document's main points and most important ideas. It should be concise and not exceed 168 words. The abstract should provide a clear and concise description of the purpose, methods, results, and conclusions of the manuscript. It should also include any relevant background information and provide a brief overview of the implications of the findings. The abstract should be written in the past tense and in a way that is accessible to a wide audience.

It is strongly recommended to add a limitations and future perspective section so that young scholars can quickly identify the future research gap and can answer it.

Author Response

ijerph-2346702: "Post-Covid symptoms in occupational cohorts: effects on health and work ability" about the COVID-19 pandemic has had a significant impact on the health and workability of many occupational cohorts. As the pandemic goes on, people are becoming more worried that people who have been infected with the virus could have long-term health effects, such as post-COVID symptoms. Post-COVID symptoms can include fatigue, shortness of breath, chest pain, and other respiratory symptoms. These symptoms can have a significant impact on an individual’s ability to perform their job duties, as well as their overall health. It’s very good research work, but there is still a weak manuscript; please improve it and follow my suggestions for clear readers.

Response: We thank the reviewer very much for his appreciation of our work and for the advice he gave us to improve the manuscript.

An abstract is a short summary of a manuscript that gives an overview of the document's main points and most important ideas. It should be concise and not exceed 168 words. The abstract should provide a clear and concise description of the purpose, methods, results, and conclusions of the manuscript. It should also include any relevant background information and provide a brief overview of the implications of the findings. The abstract should be written in the past tense and in a way that is accessible to a wide audience.

R.: We thank the reviewer for these tips. We verified that the abstract followed the recommendations of the IJERPH editorial board. We stayed within the word count limits suggested by the reviewer.

It is strongly recommended to add a limitations and future perspective section so that young scholars can quickly identify the future research gap and can answer it.

R.: Taking up the reviewer's suggestion, we separated the last two paragraphs of the discussion that referred precisely to future prospects and limitations of the work.

Reviewer 2 Report

Thank you for the opportunity to review the manuscript. Overall, a current topic for a broader readership and further exploration of this topic is certainly unique, especially to explore whether protracted symptoms were associated with occupational stress, sleep disturbances or mental health affected by the long-term consequences of COVID-19 pandemic in Italy.

A few questions / comments and suggestions:

In your discussion, one of the papers related to your research questions has not been reviewed to provide an in-depth comparison of findings across different context and timepoints. For example, this paper “Yip, Y. C., Yip, K. H., & Tsui, W. K. (2021). The transformational experience of junior nurses resulting from providing care to COVID-19 patients: From facing hurdles to achieving psychological growth. International Journal of Environmental Research and Public Health, 18(14), 7383” also relates to your study, but it was neglected. Consider acknowledging that paper and provide your insights accordingly.

In Line 259-262, what unprotected exposure, relevant to the study is not clear.

In Line 295-296, how to clearly elaborate minor manifestations of the acute illness, relevant to the study is not clear.

In Line 322-329, more elaboration of the association between occupational stress and anxiety and the number of symptoms, relevant to the study is not clear.

In Line 359-360, what is the persistent symptoms, relevant to the study is not clear.

There should be a separate section in the discussion section, stating the significance and implications of this study to the international nursing practice. The section should be specific and based on the findings.

Minor editing of English language required.

Author Response

Thank you for the opportunity to review the manuscript. Overall, a current topic for a broader readership and further exploration of this topic is certainly unique, especially to explore whether protracted symptoms were associated with occupational stress, sleep disturbances or mental health affected by the long-term consequences of COVID-19 pandemic in Italy.

A few questions / comments and suggestions:

In your discussion, one of the papers related to your research questions has not been reviewed to provide an in-depth comparison of findings across different context and timepoints. For example, this paper “Yip, Y. C., Yip, K. H., & Tsui, W. K. (2021). The transformational experience of junior nurses resulting from providing care to COVID-19 patients: From facing hurdles to achieving psychological growth. International Journal of Environmental Research and Public Health, 18(14), 7383” also relates to your study, but it was neglected. Consider acknowledging that paper and provide your insights accordingly.

Response. Thank you to the reviewer for pointing out this interesting article, a qualitative study of a group of junior nurses to examine their in-depth experience in providing care for COVID-19 patients. The article does not discuss post-Covid syndrome, so we have no way to cite it in this manuscript.

In Line 259-262, what unprotected exposure, relevant to the study is not clear.

Response: Healthcare worker exposure to Covid-19 patients without proper safety measures (distancing, use of second- or third-level masks, etc.) in the early stages of the pandemic was very common, as the paragraph preceding the quoted sentence explains. Workers who had had unprotected contact remained on duty.

In Line 295-296, how to clearly elaborate minor manifestations of the acute illness, relevant to the study is not clear.

  1. Accepting the reviewer's request, we have modified the text so that it is clearer what is referred to the literature and what comes from our own work.

In Line 322-329, more elaboration of the association between occupational stress and anxiety and the number of symptoms, relevant to the study is not clear.

Response: We can confirm what we wrote in the text, that being the cross-sectional study, we cannot say whether the symptoms cause anxiety, or whether the worker's state of anxiety makes him/her report many symptoms.

In Line 359-360, what is the persistent symptoms, relevant to the study is not clear.

R.: We changed “persistent” into “post-Covid”.

There should be a separate section in the discussion section, stating the significance and implications of this study to the international nursing practice. The section should be specific and based on the findings.

R.: Accepting the reviewer's suggestion, we have indicated in the section4.1 of the Discussion the research developments, which affect all occupational medicine and not just that aimed at nurses

Comments on the Quality of English Language. Minor editing of English language required.

R.: since none of the authors were native English speakers, we submitted the manuscript to Prof. E.A. Wright, a native English-speaking scientific translation expert.

Reviewer 3 Report

I reviewed with interest the manuscript of Magnavita et al. "Post-Covid symptoms in occupational cohorts: effects on health and work ability". In this article, the authors studied the manifestations of post-COVID syndrome in a cohort of patients examined at the workplace. A feature of this article is that not only patients who had COVID-19 were examined, but also other employees of institutions. This made it possible to compare the developed symptoms in patients who underwent COVID-19 and their colleagues. This study design allowed the authors to obtain new scientific facts that may be useful for further research and for clinicians. However, when reviewing, I had questions and comments to which I would like to receive answers from the authors.

1. First of all, noteworthy is the small number of participants in the study who had COVID-19. There were few of them (15%) even among hospital workers. I remember reporting from Italy about the beginning of the pandemic and it seemed to me that the incidence among hospital workers should be significantly higher. By the way, among hospital workers in those hospitals with which I was in contact, the percentage of cases was significantly higher (I think that it is no less than 50%). How will the authors comment on this?

2. There is no doubt that the initial severity of COVID-19 has an impact on symptoms at follow-up (1,2), including in the workplace. How did the authors of the article take this into account?

3. Occupational Status is known to affect post-COVID symptoms and cognitive function (3). Did the authors of Occupational Status take into account the individuals examined?

4. Questions on statistics. Linear regression is used to compare scores that are normally distributed. The question is whether the number of post-COVID symptoms in patients was tested for normality. Apparently the distribution was not normal, since in this case it was ordinal data.

Also, no way is specified to check other data for normality. The authors indicate that they presented quantitative data as mean and standard deviation. At the same time, Mann-Whitney's U-test (Tables 1 and 4) were used to compare groups, which are usually used in non-normal distribution. Explain why this particular test was chosen.

References:

1.     Lunt J, Hemming S, Burton K, Elander J, Baraniak A. What workers can tell us about post-COVID workability. Occup Med (Lond). 2022 Aug 15:kqac086. doi: 10.1093/occmed/kqac086. Epub ahead of print.

2.       Müller K, Poppele I, Ottiger M, Zwingmann K, Berger I, Thomas A, Wastlhuber A, Ortwein F, Schultz AL, Weghofer A, Wilhelm E, Weber RC, Meder S, Stegbauer M, Schlesinger T. Impact of Rehabilitation on Physical and Neuropsychological Health of Patients Who Acquired COVID-19 in the Workplace. Int J Environ Res Public Health. 2023 Jan 13;20(2):1468. doi: 10.3390/ijerph20021468.

3.       Delgado-Alonso C, Cuevas C, Oliver-Mas S, Díez-Cirarda M, Delgado-Álvarez A, Gil-Moreno MJ, Matías-Guiu J, Matias-Guiu JA. Fatigue and Cognitive Dysfunction Are Associated with Occupational Status in Post-COVID Syndrome. Int J Environ Res Public Health. 2022 Oct 16;19(20):13368. doi: 10.3390/ijerph192013368.

No comments

Author Response

I reviewed with interest the manuscript of Magnavita et al. "Post-Covid symptoms in occupational cohorts: effects on health and work ability". In this article, the authors studied the manifestations of post-COVID syndrome in a cohort of patients examined at the workplace. A feature of this article is that not only patients who had COVID-19 were examined, but also other employees of institutions. This made it possible to compare the developed symptoms in patients who underwent COVID-19 and their colleagues. This study design allowed the authors to obtain new scientific facts that may be useful for further research and for clinicians. However, when reviewing, I had questions and comments to which I would like to receive answers from the authors.

  1. First of all, noteworthy is the small number of participants in the study who had COVID-19. There were few of them (15%) even among hospital workers. I remember reporting from Italy about the beginning of the pandemic and it seemed to me that the incidence among hospital workers should be significantly higher. By the way, among hospital workers in those hospitals with which I was in contact, the percentage of cases was significantly higher (I think that it is no less than 50%). How will the authors comment on this?

Response: The author makes an interesting observation, which allows us to point out a relevant aspect of the study. Indeed, it is true that many health care workers to date have contracted SARS-CoV-2 infection, but one must keep in mind the time period in which this occurred. In 2022, our own longitudinal study of anesthetists in one of two hub-Covid-19 hospitals in central Italy, continuously and exclusively exposed to high risk of infection, found a 20% incidence after the first four pandemic phases; three-quarters of these infections, however, had occurred during the first wave. The incidence obviously increased as the pandemic continued. Another early meta-analysis estimated an overall incidence of 10.1% (95%CI: 5.3-14.9) of COVID-19 among HCW [Sahu]. A later meta-analysis estimated a positivity rate equal to 51.7% (95% CI 34.7-68.2), with a prevalence of hospitalization of 15.1% and mortality of 1.5%. The incidence we observed was not low, it was higher than that recorded in the first phase of the pandemic.

  1. There is no doubt that the initial severity of COVID-19 has an impact on symptoms at follow-up (1,2), including in the workplace. How did the authors of the article take this into account?

R.: We thank the reviewer for allowing us to include some interesting topics that we had left out. Taking up the reviewer's suggestion, we have added in the Results a notation on this point. The prevalence of post-Covid complaints increased in relation to the severity of the acute form: it was present in 37% of asymptomatic, 72% of mild cases, 81% of cases with home treatment, and 88% of hospitalized workers. Moreover, the duration of the acute phase also had a relationship with the occurrence of post-Covid symptoms. They were present in 55 percent of infections lasting less than one week, 64 percent of acute forms lasting up to two weeks, 76 percent of cases lasting up to three weeks, and 90 percent of longer forms.

In the Discussion, we have briefly commented these data, as follows: “In our observations, as in the literature [references], the frequency of post-Covid symptoms was correlated with the duration and severity of the acute phase.”

  1. Occupational Status is known to affect post-COVID symptoms and cognitive function (3). Did the authors of Occupational Status take into account the individuals examined?

R.: I appreciated the indication in the article that reports neuropsychological assessments conducted on 77 subjects with post-Covid syndrome from many work sectors and notes that those who did not work or were in sick leave during the examination had the greatest neuropsychological impairment. The study is interesting but has few points of contact with ours. All workers in our survey were active at work. None of them were on sick leave at the time of the survey. All were undergoing health surveillance and were fit for work. We cited the article in the Discussion, as follows: “The neuropsychological assessment of post-Covid patients showed that fatigue and cognitive dysfunction were the most frequent symptoms associated with loss of work capacity [DelgadoAlonso].

  1. Questions on statistics. Linear regression is used to compare scores that are normally distributed. The question is whether the number of post-COVID symptoms in patients was tested for normality. Apparently the distribution was not normal, since in this case it was ordinal data. Also, no way is specified to check other data for normality. The authors indicate that they presented quantitative data as mean and standard deviation. At the same time, Mann-Whitney's U-test (Tables 1 and 4) were used to compare groups, which are usually used in non-normal distribution. Explain why this particular test was chosen.

R.: As correctly the reviewer says, we ust better clarify this point. The issue of normality was not critical. As reported by Lumley et al. (2002), the assumption of normality is only required for small samples, due to the central limit theorem. With sample sizes exceeding 30, as it is the case here, violations of the normality assumptions are not problematic, and they become less and less problematic, even when extreme, as the sample size increases. We used the Mann-Whitney U test to compare the distributions of the groups because the variables investigated were ordinal and all the observations from both groups were independent of each other. We changed the section Statistics as follows: “The variables were described in terms of mean and standard deviation and checked for normality, using Kolmogorov-Smirnov and Shapiro-Wilk tests. The issue of normality was not critical in this study, because, as reported by Lumley et al. [ref], the assumption of normality is only required for small samples, due to the central limit theorem. With sample sizes exceeding 30, as it is the case here, violations of the normality assumptions are not problematic, and they become less and less problematic, even when extreme, as the sample size increases. Comparisons between the means of the variables were made using Student's t-test or Mann-Whitney's U-test, the latter for comparing ordinal variables.”

Lumley, T., Diehr, P., Emerson, S., & Chen, L. (2002). The importance of the normality assumption in large public health data sets. Annual Review of Public Health, 23(1), 151–169. https://doi.org/10.1146/annurev.publhealth.23.100901.140546

References:

Lunt J, Hemming S, Burton K, Elander J, Baraniak A. What workers can tell us about post-COVID workability. Occup Med (Lond). 2022 Aug 15:kqac086. doi: 10.1093/occmed/kqac086. Epub ahead of print.

Müller K, Poppele I, Ottiger M, Zwingmann K, Berger I, Thomas A, Wastlhuber A, Ortwein F, Schultz AL, Weghofer A, Wilhelm E, Weber RC, Meder S, Stegbauer M, Schlesinger T. Impact of Rehabilitation on Physical and Neuropsychological Health of Patients Who Acquired COVID-19 in the Workplace. Int J Environ Res Public Health. 2023 Jan 13;20(2):1468. doi: 10.3390/ijerph20021468.

Delgado-Alonso C, Cuevas C, Oliver-Mas S, Díez-Cirarda M, Delgado-Álvarez A, Gil-Moreno MJ, Matías-Guiu J, Matias-Guiu JA. Fatigue and Cognitive Dysfunction Are Associated with Occupational Status in Post-COVID Syndrome. Int J Environ Res Public Health. 2022 Oct 16;19(20):13368. doi: 10.3390/ijerph192013368.

Round 2

Reviewer 3 Report

The authors responded in detail to my comments and made the necessary corrections to the text of the manuscript. I have no other comments.

No comments